# Factors Predicting the Utilization of Center-Based Cardiac Rehabilitation Program

**DOI:** 10.3390/geriatrics5040066

**Published:** 2020-09-28

**Authors:** Lufei Young, Qi Zhang, Eric Lian, Kimberly Roberts, Neal Weintraub, Yanbin Dong, Haidong Zhu, Hongyan Xu, Pascha Schafer, Stephanie Dunlap

**Affiliations:** 1College of Nursing, Augusta University, Augusta, GA 30912, USA; 2The Nethersole School of Nursing, Faculty of Medicine, The Chinese University of Hong Kong, Hong Kong 999077, China; zhangqiphd@gmail.com; 3Departments of Medicine, Medical College of Georgia, Augusta University, Augusta, GA 30912, USA; elian@augusta.edu (E.L.); nweintraub@augusta.edu (N.W.); pschafer@augusta.edu (P.S.); sdunlap@augusta.edu (S.D.); 4Department of Nursing, School of Health Sentences, Georgia Highlands College, Rome, GA 30161, USA; kiroberts@augusta.edu; 5Vascular Biology Center, Medical College of Georgia, Augusta University, Augusta, GA 30912, USA; 6Georgia Prevention Institute, Medical College of Georgia, Augusta University, Augusta, GA 30912, USA; ydong@augusta.edu (Y.D.); hzhu@augusta.edu (H.Z.); 7Department of Biostats & Data, Medical College of Georgia, Augusta University, Augusta, GA 30912, USA; hxu@augusta.edu

**Keywords:** cardiac rehabilitation, incompletion, attendance, principal components analysis, neural network

## Abstract

Although cardiac rehabilitation (CR) is clearly beneficial to improving patients’ physical functioning and reducing heart disease progression, significant proportions of patients do not complete CR programs. To evaluate the prevalence and predictors of completion of a center-based CR program in eligible cardiac patients, existing data collected from electronic medical records were used. To identify the predictors of CR completion, we used principal components analysis (PCA) and an artificial neural network (ANN) module. Among 685 patients, 61.4% (*n* = 421) completed the program, 31.7% (*n* = 217) dropped out, and 6.9% (*n* = 47) were referred but failed to initiate the program. PCA was conducted to consolidate baseline data into three factors—(1) psychosocial factors (depression, anxiety, and quality of life), (2) age, and (3) BMI, which explained 66.8% of the total variance. The ANN model produced similar results as the PCA. Patients who completed CR sessions had greater extremity strength and flexibility, longer six-minute walk distance, more CR knowledge, and a better quality of life. The present study demonstrated that patients who were older, obese, and who had depression, anxiety, or a low quality of life were less likely to complete the CR program.

## 1. Introduction

Cardiovascular disease (CVD) is the leading cause of mortality and morbidity, and has a rising clinical and economic burden worldwide [1,2]. In management of CVD, cardiac rehabilitation (CR) is the standard treatment to promote lifestyle changes and modify risk factors for many CVD patients [3]. CR is a secondary prevention program comprised of multifaceted interventions, focused on supervised exercise training, medication management, nutritional education, psychosocial counselling, and comprehensive support [4]. Participation in CR was proven to be a beneficial element of cardiac care [5,6,7,8,9], and patients enrolled in CR exhibit reduced mortality, by up to 30% [10]. American Heart Association promotes the importance of referring appropriate patients to CR and recognizes this aspect of care in their clinical practice guidelines [11,12]. Nevertheless, only 15–30% of eligible candidates actually attend CR programs [13,14,15]. In addition, the benefits of CR are dependent on the patient’s long-term adherence to the program; among those who participate, 43% to 51% drop out within six months and up to 90% fail to complete one year [16,17]. CR incompletion rates vary in different studies and might be influenced by patient characteristics, psychological factors, social support, logistical issues, financial constraints, etc. To date, the predictors of completing a CR program are not fully understood. Therefore, the primary aim of the present study was to identify factors associated with CR completion rate among CVD patients.

## 2. Materials and Methods

### 2.1. Study Design

This study used a secondary, retrospective cohort study design. De-identified clinical data collected from the electronic medical record (EMR) were used for analysis.

### 2.2. Study Population

The de-identified medical records of 685 patients who were eligible to attend an academic medical center-based CR program in the Southeastern US, between April 2012 and April 2018 were included. The study included patients who (1) had an acute coronary syndrome; (2) underwent percutaneous coronary intervention, open-heart surgery (such as coronary artery bypass surgery, valve surgery or heart transplant); (3) had stable chronic heart failure; and (4) had current stable angina (chest pain). The patients who did not have 3-months of data available for review were excluded from the study. The study was approved by Augusta University Institutional Review Board (1295531 on 27 August 2018 and 1609080 on 27 July 2020)

### 2.3. Study Setting

The study participants were patients who were eligible for a 12-week CR program, after discharge from hospital. The CR program included three 1-h sessions per week, with a total of 36 exercise sessions. The supervised exercise training consisted of warm-up, stretching, treadmill, bicycle ergometer, and arm ergometer, with each activity performed over 10–20 min. Psychological support and nutritional counseling were offered to all patients. Blood pressure, heart rate, as well as exercise intensity were monitored by physical therapists during the CR program. Based on the completion status, we divided the participants into 3 groups—graduates, dropouts, and referrals. Graduates were patients who completed 36 CR sessions, dropouts withdrew from the program prior to completion, and referrals were sent for but failed to initiate the CR program.

### 2.4. Study Variables and Measures

In order to identify potential factors associated with CR completion, participants’ characteristics were captured at baseline, prior to the CR program and 3 months after baseline. The data included sociodemographic variables (e.g., age, gender), clinical variables (e.g., primary cardiac diagnosis, resting heart rate, blood pressure, weight, height, body mass index (BMI), percentage of body fat), laboratory variables (e.g., total cholesterol, high-density lipoproteins (HDL), triglycerides, low-density lipoproteins (LDL), fasting blood glucose (FBG), hemoglobin A1C (HbA1c)), and physical performance variables (e.g., 6-min work test (6MWT), extremity flexibility, and strength assessment). CR knowledge test [18] was used to assess patients’ knowledge regarding cardiac rehabilitation. Health-related quality of life (HRQoL) was measured by self-report, using the Short Form-36 [19] questionnaires general health (SF36GH) scales and physical functioning (SF36PF) scales. Both subscales are commonly used to assess HRQoL in cardiac patients. Higher scores in general health and physical functioning indicate better HRQoL. The Hospital Anxiety and Depression Scale [20] (HADS) was used to assess patients’ psychological symptoms. The psychometric properties of SF36 and HAD were established in a cardiac population [21,22,23].

### 2.5. Statistical Methods

All statistical analyses were performed using IBM SPSS statistics for Windows, version 26.0 (IBM Corp, Armonk, NY, USA). Data were displayed as mean ± SD or percentage (N) for numerical variables or categorical variables, respectively. For all analyses, two-tailed *t*-tests were used with a level of statistical significance set at *p* < 0.05. The student’s paired *t*-test was used to examine differences before and after, for each group. One-way analysis of variance (ANOVA) was employed to identify the relationships between variables.

Principal components analysis (PCA), a statistical procedure aimed to reduce the dimensionality of multiple correlated measurements to fewer numbers of linearly uncorrelated variables, was used to investigate the variation of the original data [24,25]. PCA can highlight the common variation between the original variables to condense the data, thus, identifying the most relevant principal components to explain CR utilization status (Graduate, Dropout, and Referral). Each principal component (PC) is, by definition, not related to another. The first PC (PC1) obtained accounts for the highest amount of the total variation between the original explanatory variables, while the next components (PC2, PC3…), respectively, accounted for less variation. Moreover, Kaiser–Meyer–Olkin (KMO) and the Bartlett test of sphericity was performed to verify the appropriateness of the sample, a varimax rotation was used to improve the interpretability of performing PCA, and the cut-off point was a variable loading ≥ 0.6.

Artificial neural network (ANN) modules are non-linear mapping structures that imitate the learning process of the human brain. They are powerful tools for processing problems involving non-linear and complex data, especially when the underlying data relationship is imprecise and noisy. Variables that are identified by linear modeling to be statistically associated with CR utilization status (Graduate, Dropout, and Referral) were used for neural network modeling. The data set included three distinct sets—training, testing, and validation. The training set was used to learn data patterns, the testing set was to evaluate the generalization ability of the trained network, and the validation set was used to check the trained network performance. The ANN procedure was carried out using a PC laptop computer, equipped with a 64-bit operating system, 2.90 GHz microprocessor, and 8.00 GB of RAM.

In this study, we used clinical data to identify the predictive model of CR utilization, which posed significant challenges for statistical analysis. The clinical data collected over time were voluminous, with large number of variables that were correlated or uncorrelated. There were many different types of observations with different levels of measure and measurement scales, which made it difficult to standardize. Large number of missing data reduced power and validity. The temporal relationships between variable were often nonlinear [26]. PCA is recommended as an alternative to classical regression analysis of multivariable clinical data [27,28]. However, the drawback of PCA is that it only recognizes the linear relationships, which limits its ability to identify nonlinear relationships between clinical variables [28,29], as the main strength of ANN is to handle nonlinear and multidimensional dependencies [29,30]. However, to identify the best-fit predictive model, ANN can be labor-intensive and time consuming, when dealing with a large number of clinical variables embedded in multi-layer and multi-level data structures [30]. To improve efficiency, PCA was first performed to reduce the input dimensions before ANN. PCA also helped develop ANN training sets and enhance the specificity of a predictive model by reducing dimensionality and removing the correlated variables. Then, ANN used the principal components obtained by the PCA method to develop a better performed predictive model with higher accuracy [28,29,30]. Using both PCA and ANN could help address challenges in clinical data analysis and interpretation. The combination of PCA and ANN is a valid and effective approach to reduce residual errors, make proper classification, recognize relationship patterns, and test an unlimited number of related or unrelated factors among a large number of clinical observations [28,29,30].

## 3. Results

### 3.1. Baseline Characteristics

The baseline demographic and clinical characteristics of the study population is summarized in Table 1. Of the 685 patients included in the study, 421 (61.4%) of them completed 36 sessions, 217 (31.7%) dropped out, and 47 (6.9%) were referred to by a physician but failed to initiate the program. The average age of the study population was 64 years, ranging from 24 to 89. Male patients accounted for 65% of the study population. A total of 42% underwent coronary artery bypass grafting (CABG) or percutaneous coronary intervention, 28% had a stable angina, 20% had acute coronary syndrome, and 8% had a stable heart failure.

### 3.2. Comparison between Baseline and 3 Months

Comparison analyses were performed in the three groups (Graduates, Dropouts, and Referrals) at baseline and at 3 months (the standard duration of our program). Graduates exhibited greater extremity strength and flexibility, longer six-minute walk distance (6MWT), superior CR knowledge score, and a higher SF-36 PF score (all *p* < 0.001 by paired *t*-test). Moreover, the Graduates had significantly reduced anxiety and depression compared to Dropouts and Referrals, after completing CR (*p* = 0.000, paired *t*-test). While both Dropouts and Referrals had increased 6MWT distance (*p* = 0.038 and *p* = 0.001) and 6MWT Metabolic equivalent of tasks (METs) (*p* = 0.038 and *p* = 0.001), respectively, after 3 months, significant changes were not observed in the other variables (Figure 1 and Figure 2, Appendix A).

### 3.3. Predictors of CR Completion

One-way ANOVA test was used to analyze the demographic factors contributing to the CR completion status. According to the results presented in Table 2, there was a statistically significant difference between the three groups regarding mean age, BMI, CR knowledge test, HADS Anxiety, HADS Depression, SF36GH, and PF (*p* < 0.05).

Principal component analysis (PCA) was conducted to consolidate baseline data into three factors, which explained 66.8% of the total variance. The Kaiser-Meyer-Olkin (KMO) statistic was 0.75, which exceeded the minimum recommendation of 0.60, and the Bartlett test of sphericity was statistically significant (χ^2^ = 267.885, *p* < 0.001), both indicating that the sampling adequacy and correlation matrix was satisfactory for PCA.

The first principal components (PC1) addressed psychosocial variables (including SF36GH, HADS Depression, HADS Anxiety, and SF36PF), which accounted for 38.5% of the variance. The second principal component (PC2), which only included age as a variable, contributed to 14.5% of the variation. Finally, the third principal component (PC3), which included BMI variables, explained an additional 13.8% of the variation. PCA showed that patients who were older, obese, and who had depression, anxiety, or a low quality of life, were less likely to complete the CR program. A summary of the loadings, after varimax rotation, related to the variables in each component, is provided in Table 3. Appendix A shows the scree plot of the three components. A detailed analysis of the score of each observation on the three PC analyses, as well as the contributing effect of each variable, are presented in Figure 3.

Finally, ANN was used to validate the results from PCA. It used feed-forward architecture with multi-layered perception. The same seven variables identified by linear modeling were confirmed in the input layer of the ANN module (Table 4). The description diagram of the ANN model, generated by the software iterations, is presented in Figure 4. Our results identified the input layer comprised of seven neurons, which validated the findings generated from PCA. The output layer consisted of three neurons, representing three groups, based on CR utilization status. The hidden layer located between the input and output layers, contained mathematical functions that performed nonlinear transformations of the input entered into the network. The percent incorrect predictions for the training and testing steps were 0.197 and 0.229, respectively (Table 5), which was acceptable. The algorithm for determination of predictor importance, also termed “specificity,” is presented in Appendix A. After the nonlinear transformation in the hidden layer and two-step modeling (training and testing), ANN was able to classify all study subjects into three groups, based on the minimal number of predictors needed, as they were entered into the input layer of the network.

## 4. Discussion

This study aimed to identify the predictors of CR completion among patients with CVD. Among patients who were eligible for CR, approximately 32% of them dropped out, and 7% referred patients failed to initiate CR therapy. Identified predictors of completion fall into three categories—psychosocial, demographic, and clinical aspects, which account for 66.8% of the total variation among the studied variables.

The psychosocial domain included HRQoL (assessed by SF-36 PF and SFGH), anxiety, and depression (assessed by HADS). Consistent with previous studies, HRQoL played a significant role in patients’ decision to utilize CR. Compared to patients who completed CR, those who failed to initiate or complete the CR program had lower HRQoL at baseline. Most studies reported an improvement of HRQoL after CR completion, which was congruent with our findings, but few studies examined the impact of HRQoL on CR utilization in CVD patients [31]. Unlike graduates, who had significantly improved the quality of life after completing CR, those who dropped out or who were referred but did not attend CR showed no significant improvement in QoL over time, after hospital discharge, suggesting that this is a critical window of time for patients to attend CR. In addition, HRQoL is related to psychological anxiety and depression [32], which was significantly improved in those who completed CR. Interestingly, our study also showed that anxiety and depression predicted CR completion. Thus, CVD patients need to improve their quality of life and reduce their anxiety and depression, which in turn might lead to higher CR adherence. This finding was in accordance with recent clinical studies [33].

As reported in previous studies conducted by Lane et al. [8] and Petrie et al. [34], age was the only factor in the demographic domain that was significantly associated with underutilization of CR. The potential explanations included physical constraints, health status, lack of transportation, and insufficient family support. Older cardiac patients were also more likely to be single, widowed, or to live alone with less social support, leading to the underutilization of CR. Additionally, the average number of comorbidities increased monotonically with age. These comorbidities might hamper participation in exercise sessions and create barriers to completing the CR program.

Among all variables examined in the clinical domain, BMI was found to predict CR completion. Recent studies provided strong evidence that obesity is associated with physical inactivity [35]. Physically inactive patients are more likely to drop out or fail to initiate a CR program [35]. The majority of participants in our study were overweight or obese (the average BMI was 31). The average BMI was higher in the Dropout and Referral groups than the Graduate group, at baseline (34, 32, and 30, respectively). CR programs could help obese patients achieve better health outcomes by managing their blood pressure, diabetes, and lipid levels. The effect of CR was dose-dependent, which meant that the number of sessions attended was positively associated with weight loss and reductions in blood pressure. Therefore, additional studies are needed to identify factors influencing obese patients’ decision making on CR utilization, thus, enabling development of a tailored intervention to promote CR utilization in obese cardiac patients.

There are several limitations in our study. First, this study was an observational investigation using a convenient sample of patient data from a single CR program in an academic, hospital-based setting. Therefore, the results might not be generalized to national or international CR patients. Based on a recent study describing national CR participants’ characteristics [36], however, our study included a heterogeneous and representative sample, comparable to national CR patients. In addition, the advanced statistical modeling helped reduce bias and improve generalizability to other CR programs across the nation. We used the combination of principal component analysis (PCA) and artificial neural network (ANN) analysis to examine predictors of both CR completion and attendance. PCA detects the common variation between the original variables and condenses the data. In the present study, a large number of explanatory variables makes PCA reliable for the purpose of the study. ANN was able to detect non-linear relationships that were not identified by PCA. The combination of PCA and ANN is a valid and effective approach to enhance model predictive accuracy, when dealing with complex clinical data with large missing values.

Second, the list of explanatory variables that could contribute to CR utilization was limited to what was available in the EMR. Factors such as economic scarcity, financial scarcity, social support, health belief, health literacy, and personality traits, were not available in EMR [37]. These missing variables could lead to misleading conclusions regarding the relative importance of the characteristics examined here. On the other hand, the selected variables were captured by most CR programs nationwide and used to develop clinically feasible interventions to promote CR utilization. Third, the HRQoL and mental health assessments were obtained from patients’ self-reports, which was not validated by objective measures and thus could have led to bias. To date, objective and clinically feasible measures of HRQoL, anxiety, and depression are not identified or reported. As a result, future studies are needed to replicate and expand our findings.

## 5. Conclusions

The study identified important factors predicting CR utilization. Our statistical modeling revealed three domains that explained 66.8% of the total variance, including age, BMI, HRQoL, anxiety, and depression. Pre-CR screening and tailored interventions are needed to promote CR utilization in high-risk populations.

## Figures and Tables

**Figure 1 geriatrics-05-00066-f001:**
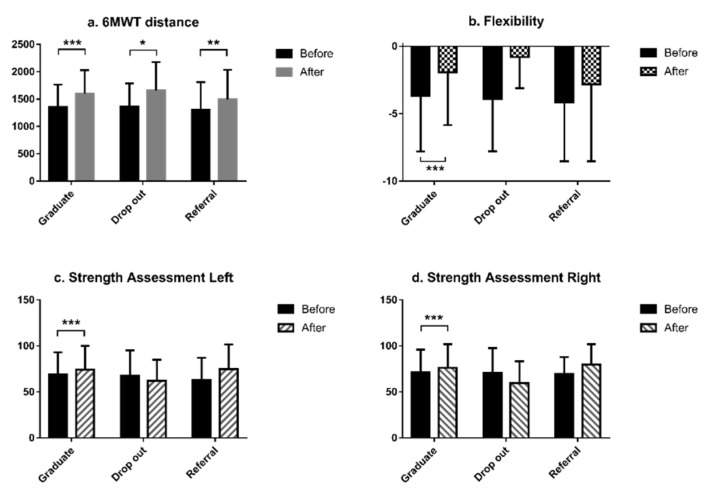
Physical assessment before and after cardiac rehabilitation in the three patient groups (Graduate, Dropout, and Referral). (**a**) Six-minute walk test (6MWT) distance differences in paired *t*-tests; (**b**) flexibility differences in paired *t*-tests; (**c**) strength assessment of left arm differences in paired *t*-tests; and (**d**) strength assessment of right arm differences in paired *t*-tests. * *p* < 0.05, ** *p* < 0.01, and *** *p* < 0.001.

**Figure 2 geriatrics-05-00066-f002:**
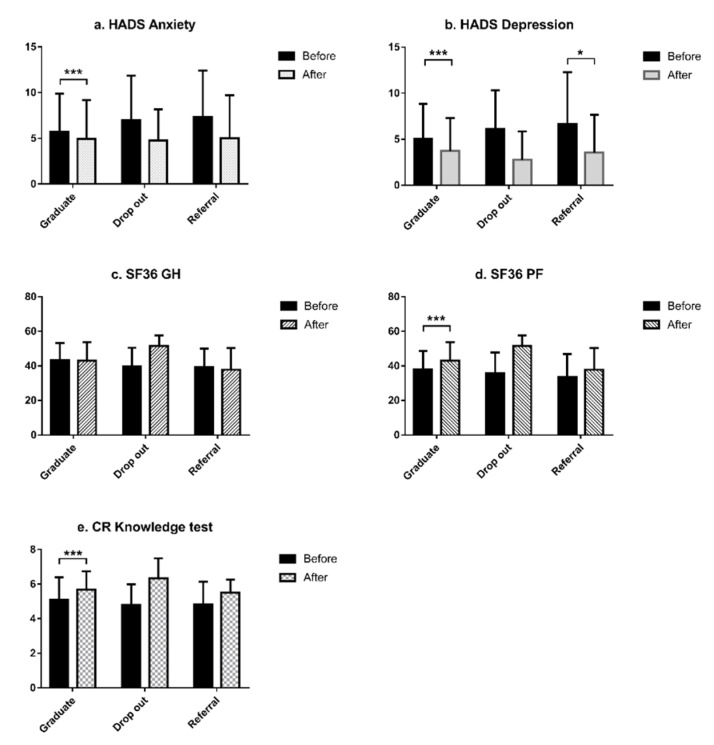
Psychosocial assessment before versus after cardiac rehabilitation in the three patient groups. (**a**) Hospital anxiety and depression scale (HADS) anxiety differences in paired t tests; (**b**) HADS depression differences in paired t tests; (**c**) Short Form 36 for general health (SF36GH) differences in paired t tests; (**d**) Short Form 36 for physical functioning (SF36PF) differences in paired t tests; and (**e**) cardiac rehabilitation (CR) knowledge test differences in paired *t*-tests. * *p* < 005, *** *p* < 0.001.

**Figure 3 geriatrics-05-00066-f003:**
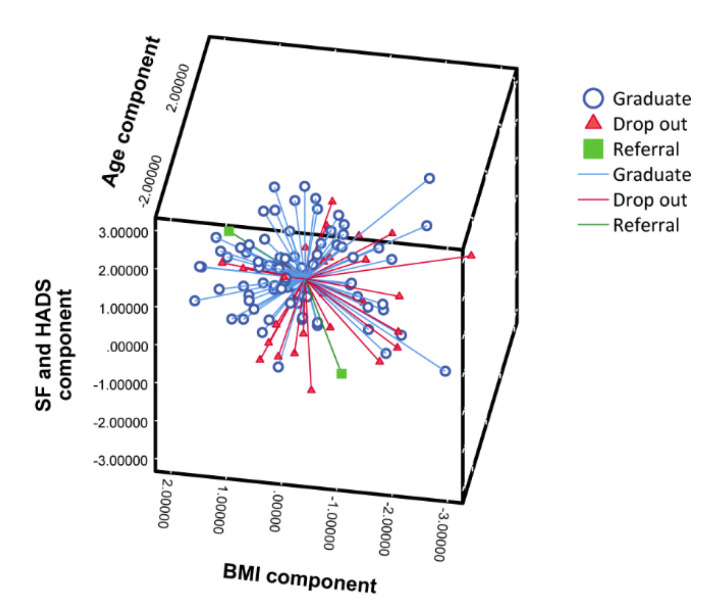
3D-plot indicating the score of each observation on the three principal components, as well as the contributing effect of each variable (expressed as the product of loadings and singular values). HADS, hospital anxiety and depression scale; SF, Short Form; and BMI, body mass index.

**Figure 4 geriatrics-05-00066-f004:**
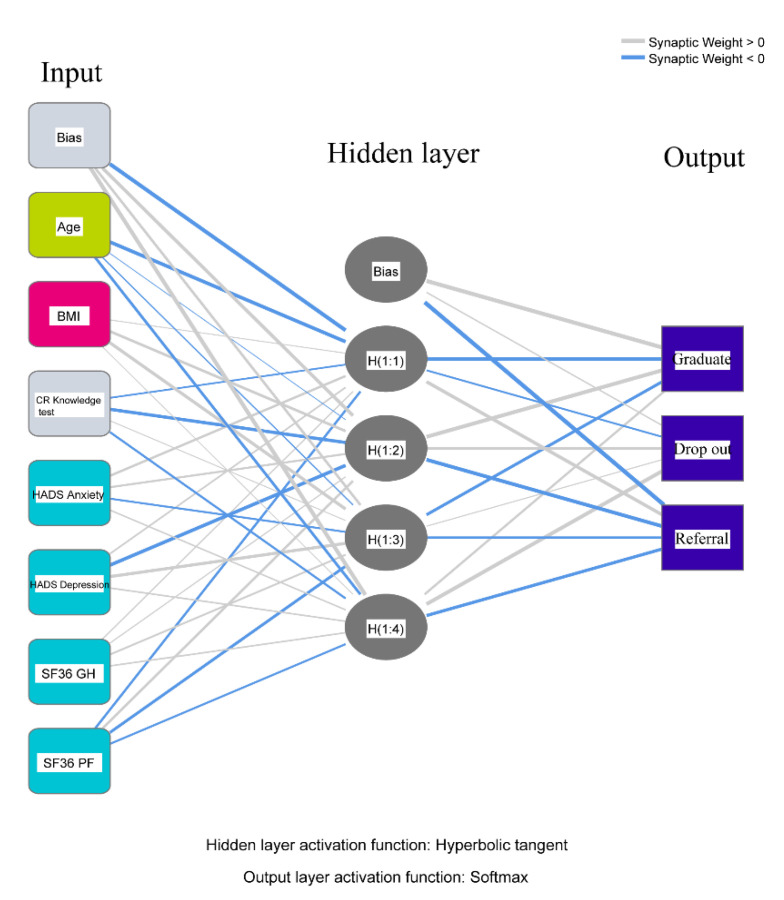
The architecture of the typical neural network utilized in the artificial neural network module. BMI, body mass index; CR, cardiac rehabilitation; HADS, hospital anxiety and depression scale; SF, Short Form; GH, general health; and PF, physical functioning.

**Table 1 geriatrics-05-00066-t001:** Baseline patient characteristics across cardiac rehabilitation status categories.

Characteristic		Total	Graduate	Drop-Out	Referral
		Mean ± SD or N (%)	Mean ± SD or N (%)	Mean ± SD or N (%)	Mean ± SD or N (%)
Status (Graduate, Drop out, Referral)		685 (100%)	421 (61.4)	217 (31.7)	47 (6.9)
Age		64.0 ± 12.5	65.8 ± 11.7	60.9 ± 13.0	69.4 ± 14.3
Gender	male	443 (64.7)	274 (65.1)	133 (61.3)	36 (76.6)
	female	242 (35.3)	147 (34.9)	84 (38.7)	11 (23.4)
Primary diagnosis	Arrhythmia	3 (0.4)	2 (0.5)	1 (0.5)	
	Stable angina	189 (27.6)	115 (27.4)	84 (38.7)	17 (36.1)
	Acute coronary syndrome	135 (19.7)	76 (18.1)	49 (22.6)	10 (21.3)
	Stable heart failure	54 (7.9)	33 (7.8)	18 (8.3)	3 (6.4)
	CABG or PTCA	288 (42.0)	187 (44.4)	57 (26.3)	17 (36.2)
	Other	16 (2.3)	8 (1.9)	8 (3.7)	
Resting HR		71.9 ± 12.1	71.4 ± 11.8	72.9 ± 12.8	73.6 ± 12.4
SBP		127.1 ± 20.7	127.4 ± 19.8	126.8 ± 23.1	125.8 ± 16.9
DBP		72.0 ± 12.3	72.7 ± 12.1	70.8 ± 13.1	71.1 ± 10.4
TC		162.3 ± 49.4	159.9 ± 46.1	165.3 ± 54.2	162.0 ± 42.6
HDL		42.5 ± 13.9	43.4 ± 13.3	41.2 ± 13.3	43.43 ± 20.8
Trig		163.1 ± 272.0	139.2 ± 95.7	190.0 ± 401.1	177.8 ± 157.2
LDL		91.5 ± 40.6	91.9 ± 40.5	91.7 ± 41.9	87.9 ± 33.0
FBG		123.4 ± 93.2	123.8 ± 107.7	123.2 ± 63.3	121.4 ± 40.3
Hemoglobin A1c		6.9 ± 6.0	7.0 ± 7.6	6.7 ± 1.9	6.8 ± 1.7
Weight		201.7 ± 49.6	200.4 ± 49.8	203.2 ± 49.8	206.8 ± 47.5
Height		67.6 ± 4.2	68.0 ± 3.8	67.1 ± 4.4	66.6 ± 5.7
BMI		31.1 ± 8.0	30.4 ± 6.8	31.8 ± 8.0	34.0 ± 15.2
%BF		34.1 ± 8.2	33.7 ± 8.4	34.7 ± 7.9	36.8 ± 6.1

SD, standard deviation; CABG, Coronary artery bypass grafting; PTCA, Percutaneous transluminal coronary angioplasty; HR, heart rate; SBP, systolic blood pressure; DBP, diastolic blood pressure; TC, Total Cholesterol; HDL, High-density lipoprotein; Trig, Triglycerides; LDL, Low-density lipoprotein; FBG, Fasting blood glucose; BMI, body mass index; and % BF, Percentage of body fat.

**Table 2 geriatrics-05-00066-t002:** One-way ANOVA factors contributing to the status groups.

Variables	F	*p*-Value
Age	4.369	0.014 *
6MWT distance	0.216	0.806
6MWT Mets	0.220	0.803
Resting heart rate	1.220	0.296
Systolic BP	0.149	0.861
Diastolic BP	1.821	0.163
Limb flexibility	0.276	0.759
Left arm strength	0.839	0.433
Right arm strength	0.047	0.954
BMI	5.117	0.006 **
% of body fat	2.564	0.078
CR Knowledge test	3.622	0.027 *
HADS Anxiety	4.381	0.013 *
HADS Depression	4.137	0.017 *
SF36GH	6.717	0.001 ***
SF36PF	2.966	0.052

* *p* < 0.05, ** *p* < 0.01, and *** *p* < 0.005 F, F Statistics; 6MWT, six-minute walk; BP, blood pressure; BMI, body mass index; HADS, hospital anxiety and depression scale; SF, Short Form; GH, general health; and PF, physical functioning.

**Table 3 geriatrics-05-00066-t003:** Variable loading matrix and explained variance related to each PC after varimax rotation.

Variable	Component		
	PC1	PC2	PC3
SF36 GH	0.872	−0.094	0.066
HADS Depression	−0.852 *	−0.030	−0.057
HADS Anxiety	−0.808 *	0.187	−0.037
SF36 PF	0.761 *	0.342	0.145
Age	0.341	−0.681 *	0.325
BMI	−0.213	0.096	−0.835 *
CR Knowledge test	−0.362	0.469	0.516

Extraction Method: Principal Component Analysis. Rotation Method: Varimax with Kaiser Normalization. Rotation converged in 7 iterations; 3 components were extracted. * A factor loading of one independent variable was considered as large if its absolute value ≥ 0.6. PC, Principal component; SF, Short Form; GH, general health; HADS, hospital anxiety and depression scale; PF, physical functioning; BMI, body mass index; and CR, cardiac rehabilitation.

**Table 4 geriatrics-05-00066-t004:** ANN parameter estimates.

Predictor		Predicted						
		Hidden Layer 1				Output Layer		
		H(1:1)	H(1:2)	H(1:3)	H(1:4)	Graduate	Drop Out	Referral
Input Layer	(Bias)	−0.991	0.508	0.351	1.082			
	Age	−0.669	−0.014	−0.094	−0.393			
	BMI	0.038	0.353	0.459	0.016			
	CR Knowledge test	−0.165	−0.504	0.030	−0.247			
	HADS Anxiety	0.248	0.225	−0.215	0.110			
	HADS Depression	0.157	−0.639	0.437	0.143			
	SF36GH	0.096	0.081	0.184	0.134			
	SF36PF	−0.270	0.285	−0.402	−0.233			
Hidden Layer 1	(Bias)					1.239	0.097	−1.422
	H(1:1)					−0.751	−0.189	0.609
	H(1:2)					0.885	0.402	−0.740
	H(1:3)					−0.413	0.069	−0.329
	H(1:4)					0.248	0.853	−0.461

The prediction weights generated by the neural network for each interaction among the 7 pre-incision factors (“input layer”) and the 4 nodes (“hidden layer”), and the output weights of each node to the prediction of CR status—bias weights, were also contributed from the input layer and the hidden layer.

**Table 5 geriatrics-05-00066-t005:** Model summary of the artificial neural network.

**Training Model Summary**	**Value**
Cross Entropy Error	42.709
Percent Incorrect Predictions	19.7%
Stopping Rule Used	1 consecutive step(s) with no decrease in error ^a^
Training Time	0:00:00.02
**Testing Model Summary**	**Value**
Cross Entropy Error	19.417
Percent Incorrect Predictions	22.9%

Dependent Variable: Status (Graduate, Dropout, and Referral). ^a^, error computations are based on the testing sample. This model displayed a summary of the neural network results by partition, and overall, including the sum of squares errors in the training and testing groups, the relative errors in the training and testing groups, the stopping rule used to stop training, and the training time.

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
