# Peer review of "Factors Predicting the Utilization of Center-Based Cardiac Rehabilitation Program"

_geriatrics, 2020, doi:10.3390/geriatrics5040066_

Round 1
Reviewer 1 Report
In this study, authors evaluated predictor of cardiac rehabilitation program utilization in a study population of 685 patients.
Psychosocial factors, age and body mass index were able to identify patients who completed the programme. Furthermore well known benefits of cardiac rehabilitation (increase in extremity strength, distance walked at six minute walking test, and quality of life) were confirmed.
The study is interesting and well written; statistical methodology seems appropriate. The major limitation Is that it is a single centre study, so local factors could explain the results of the study.
I think it would be interesting to know (if available) the number of patients not referred at all for cardiac rehabilitation.
Reviewer 2 Report
This study sought to identify predictors of completion of centre-based cardiac rehabilitation (CR) using electronic medical record data and two methods of analysis: principal components analysis (PCA) and Artificial Neural Network (ANN) analysis.
Overall, this paper is well written and answers an important question about CR. I have a small number of suggestions for the authors:
It would be useful to add to the references listed as 5 to 8, a reference related to CR attendance effect on mortality:
Beauchamp, A., et al. (2013). "Attendance at cardiac rehabilitation is associated with lower all-cause mortality after 14 years of follow-up." Heart 99: 620-625.
Lines 82 to 86: It is also useful to note some references where it is stated that both the SF36 and HADS have been validated in cardiac populations, for example:
Failde, I. and I. Ramos (2000). "Validity and reliability of the SF-36 Health Survey Questionnaire in patients with coronary artery disease." Journal of Clinical Epidemiology 53(4): 359-365.
Hunt-Shanks, T., et al. (2010). "A psychometric evaluation of the Hospital Anxiety and Depression Scale in cardiac patients: Addressing factor structure and gender invariance." British Journal of Health Psychology 15(1): 97-114.
Stafford, L., et al. (2007). " Validity of the Hospital Anxiety and Depression Scale and patient health questionnaire-9 to screen for depression in patients with coronary artery disease." Gen Hosp Psychiatry 29: 417-424.
Lines 105 to 113. It would be worth considering giving some more detail on what the ANN modules add to PCA.
Lines 182 to 202 and Figure 4. These results need more explanation for readers not familiar with this form of analysis eg what is the meaning of the hidden layer? Basically, how are these data to be interpreted? The PCA results are incorporated into the discussion, but no mention is made of the ANN.
I would like to see health literacy added to the list of factors not available in the data set analysed in lines 251 to 253, as this has been shown to be a factor in CR completion, eg Beauchamp, A., et al. (2020). "Health Literacy of Patients Attending Cardiac Rehabilitation." Journal of Cardiopulmonary Rehabilitation and Prevention 40(4).
